A new leptoceratopsid dinosaur from Maastrichtian-aged deposits of the Sustut Basin, northern British Columbia, Canada

http://orcid.org/0000-0003-4788-0546 Arbour Victoria M. 1 varbour@royalbcmuseum.bc.ca
Evans David C. 2
1 Department of Knowledge, Royal BC Museum , Victoria, BC , Canada
2 Department of Natural History, Royal Ontario Museum , Toronto, ON , Canada
Sues Hans-Dieter
Electronic publication date: 2019 Nov 7
Publication date: 2019
Volume: 7
Electronic Location ID: e7926
Received 2019 Jun 24; Accepted 2019 Sep 20
Copyright: © 2019 Arbour and Evans
Copyright year: 2019
Copyright holder: Arbour and Evans
License: This is an open access article distributed under the terms of the Creative Commons Attribution License, which permits unrestricted use, distribution, reproduction and adaptation in any medium and for any purpose provided that it is properly attributed. For attribution, the original author(s), title, publication source (PeerJ) and either DOI or URL of the article must be cited.
License URL: https://creativecommons.org/licenses/by/4.0/

Keywords: Dinosauria, Ceratopsia, Cretaceous, Biogeography, Intermontane terrane, Sustut Basin

Funding: NSERC postdoctoral fellowship, an NSERC L’Óreal-UNESCO for Women in Science fellowship supplement, a National Geographic Society Waitt Grant, and a Dinosaur Research Institute grant to Victoria Arbour, and an NSERC Discovery Grant to David Evans Funding for this project was provided by an NSERC postdoctoral fellowship, an NSERC L’Óreal-UNESCO for Women in Science fellowship supplement, a National Geographic Society Waitt Grant, and a Dinosaur Research Institute grant to Victoria Arbour, and an NSERC Discovery Grant to David Evans. The funders had no role in study design, data collection and analysis, decision to publish, or preparation of the manuscript.

==============================
A partial dinosaur skeleton from the Sustut Basin of northern British Columbia, Canada, previously described as an indeterminate neornithischian, is here reinterpreted as a leptoceratopsid ceratopsian, Ferrisaurus sustutensis, gen. et. sp. nov. The skeleton includes parts of the pectoral girdles, left forelimb, left hindlimb, and right pes. It can be distinguished from other named leptoceratopsids based on the proportions of the ulna and pedal phalanges. This is the first unique dinosaur species reported from British Columbia, and can be placed within a reasonably resolved phylogenetic context, with Ferrisaurus recovered as more closely related to Leptoceratops than Montanoceratops. At 68.2–67.2 Ma in age, Ferrisaurus falls between, and slightly overlaps with, both Montanoceratops and Leptoceratops, and represents a western range extension for Laramidian leptoceratopsids.

Introduction

The dense boreal forest and thrusted, folded rocks of the Canadian Cordillera present a challenging environment in which to search for dinosaurs, compared to the better exposed and more easily accessible outcrops in the badlands of the prairie provinces. Nevertheless, a dinosaur specimen (RBCM P900) consisting of articulated and disarticulated limb and girdle elements was discovered in 1971 in the remote interior mountains of north-central British Columbia (Fig. 1; Arbour & Graves, 2008). These bones were collected by geologist Kenny F. Larsen, who was surveying for uranium along the then in-construction BC Rail line along the Sustut River, and were later donated to Dalhousie University (Halifax, NS) and subsequently accessioned at the Royal British Columbia Museum in Victoria, BC. Arbour & Graves (2008) described this material and identified it as an indeterminate small-bodied, bipedal neornithischian, possibly representing either a pachycephalosaur or a basal ornithopod similar to Thescelosaurus. Here, we provide a new interpretation of this material and argue for its assignment to Leptoceratopsidae as a new genus and species. Leptoceratopsids were short-frilled, hornless ceratopsians with a maximum body length of about two-to-three meters, and form the sister group to all other coronosaurian neoceratopsians (He et al., 2015). They were present in many Campanian–Maastrictian aged dinosaur assemblages from Asia and North America, but are generally rare in the fossil record (Ryan et al., 2012; Longrich, 2016).

Figure 1 RBCM P900, the holotype of Ferrisaurus sustutensis, was collected along the BC Rail line near the intersection of Birdflat Creek and the Sustut River in 1971, in the Sustut Basin of northern British Columbia, Canada.

Map modified from Evenchick et al. (2003).

RBCM P900 is one of the only vertebrate fossils yet described from the Sustut Basin and as such is significant for understanding the distribution and evolution of dinosaurs in western North America. A 2017 survey of the field area near the confluence of Birdflat Creek and the Sustut River recovered a fragment of the Cretaceous turtle Basilemys at a location closely matching Larsen’s original field notes, suggesting that RBCM P900 most likely derived from the same outcrop (Fig 1; Arbour et al. (in press)). This work generated new stratigraphic and palynological data that allows the provenance of this important skeleton to be documented in detail for the first time. RBCM P900 is likely from the Tango Creek Formation, rather than the Brothers Peak Formation as originally reported by Arbour & Graves (2008), and the new palynological data suggest that the specimen is late Maastrichtian in age, allowing its morphology and biogeography to be understood in a more detailed temporal context.

Methods

The electronic version of this article in portable document format will represent a published work according to the International Commission on Zoological Nomenclature (ICZN), and hence the new names contained in the electronic version are effectively published under that Code from the electronic edition alone. This published work and the nomenclatural acts it contains have been registered in ZooBank, the online registration system for the ICZN. The ZooBank Life Science Identifiers (LSIDs) can be resolved and the associated information viewed through any standard web browser by appending the LSID to the prefix http://zoobank.org/. The LSID for this publication is: urn:lsid:zoobank.org:pub:D1C60A34-3632-43AD-BCE0-C93D5E26D1B0. The online version of this work is archived and available from the following digital repositories: PeerJ, PubMed Central and CLOCKSS. No permits were required for this study and all fossils are permanently accessioned in repositories.

RBCM P900 was compared to ceratopsian, pachycephalosaurid, ornithopodan, and parksosaurid dinosaurs in various collections (Supplemental Information 1) and the literature, and comparative measurements are provided in Supplemental Information 2. Photogrammetric digital models of the specimen (Supplemental Information 3) were created using Agisoft Metashape 1.5.4 using between 50 and 200 digital photos (in RAW format, converted to TIFFs) taken with a Canon Rebel XTi.

We assessed the phylogenetic position of RBCM P900 using the character-taxon matrix for ceratopsians presented by He et al. (2015), derived from previous matrices built by Farke et al. (2014), Ryan et al. (2012), and Makovicky (2001). Our matrix includes 34 taxa and 165 characters (Supplemental Informations 1 and 4) and was compiled in Mesquite v3.04 build 725 (Maddison & Maddison, 2011). We added three new characters (characters 163–165) based on observations made over the course of this study. We also tested the position of RBCM P900 using the character-taxon matrix presented by Morschhauser et al. (2019), with no modifications to the matrix other than the addition of RBCM P900. We performed a cladistic parsimony analysis on both matrices using the Traditional Search option in TNT v1.5 (Goloboff, Farris & Nixon, 2008); all characters were treated as unordered and of equal weight, and we used the tree bisection reconnection swapping algorithm with 1,000 replications.

Systematic palaeontology

DINOSAURIA Owen, 1842

ORNITHISCHIA Seeley, 1888

NEORNITHSICHIA Cooper, 1985

MARGINOCEPHALIA Sereno, 1986

CERATOPSIA Marsh, 1890

NEOCERATOPSIA Sereno, 1986

CORONOSAURIA Sereno, 1986

LEPTOCERATOPSIDAE Nopcsa, 1923

FERRISAURUS SUSTUTENSIS gen. et sp. nov. urn:lsid:zoobank.org:act:A7F4267C-8CC6-49B6-8E52-2C2148929B14

Diagnosis: Ferrisaurus can be differentiated from other known leptoceratopsids based on the following unique combination of characters: penultimate pedal phalanges in digits III and IV are equal or subequal in proximodistal length compared to the length of the preceding phalanx, rather than shorter as in all other leptoceratopsids for which these elements are preserved except possibly USNM 13863 (Cerasinops); astragalus and tibia coossified, unlike all other leptoceratopsids except for AMNH 5464 (Montanoceratops); distal end of ulna broader relative to radius length than in Leptoceratops; distal end of ulna medially bowed, unlike the straight ulna of the penecontemporaneous Maastrichtian taxa Leptoceratops and Montanoceratops, but similar to Cerasinops and Prenoceratops from the Campanian.

Etymology: “Iron lizard,” from Latin ferrum (=iron) and Greek sauros (=lizard), in reference to the specimen’s discovery along a railway line, and sustutensis in reference to its provenance near the Sustut River and within the Sustut Basin.

Holotype: RBCM P900, a partial skeleton consisting of a partial right coracoid, fragmentary left scapula, complete left radius, distal portion of the left ulna, associated distal two thirds of the left tibia and fibula and coossified astraglus and ?calcaneum, partial articulated digits III and IV of the right pes, and an unprepared block removed from the posterior surface of the tibia that appears to contain four metatarsals, presumably from the left pes. Previously catalogued as RBCM.EH2006.019.0001 to RBCM.EH2006.019.010 and published under RBCM.EH2006.019 by Arbour & Graves (2008).

Locality: RBCM P900 was discovered near the confluence of Birdflat Creek and the Sustut River in the Sustut Basin (Fig. 1); the bones were found loose in the rubble during construction along the BC Rail line, which has since been abandoned. Fieldwork in the Sustut Basin in 2017 provided strong support for the relocation of the original collection site a few hundred meters from the confluence of the Sustut River and Birdflat Creek (Arbour et al. (in press)); exact GPS coordinates are on file at the Royal BC Museum.

Formation and Age: Tatlatui Member, Tango Creek Formation, Sustut Group. Palynomorphs recovered from the presumed holotype locality included the Maastrichtian marker taxon Pseudoaquilapollenites bertillonites, indicating an age of approximately 68.2–67.2 Ma for the site (Arbour et al. (in press)).LSID: urn:lsid:zoobank.org:act:A7F4267C-8CC6-49B6-8E52-2C2148929B14

Description and comparison

RBCM P900 includes multiple elements in articulation, including the tibia and fibula, several pedal phalanges, and potentially the metatarsals (Fig. 2). The presence of metatarsals in a block of sediment removed from the posterior face of the distal tibia suggests that the skeleton may have been fully articulated in situ. The bones do not appear to have suffered from brittle or plastic deformation, but they have been recrystallized, obscuring the original internal bone textures.

Figure 2 Preserved elements of RBCM P900, holotype of Ferrisaurus sustutensis, in white (gray represents missing parts of incomplete bones).

RBCM P900 includes a partial right coracoid, partial left scapular blade, complete left radius, partial left ulna, partial left tibia, fibula, and coossified astragalus and ?calcaneum, partial left metatarsals I-IV, and digits III (phalanges 2–4) and IV (phalanges 2–5) of the right pes.

We reinterpret RBCM P900 as a leptoceratopsid based on several aspects of the preserved phalanges. The non-ungual phalanges are blockier and more robust in comparison to most orodromines (e.g., Orodromeus MOR 623B), parksosaurids (e.g., Parksosaurus ROM 804), and pachycephalosaurids (e.g., Stegoceras UALVP 2). The dorsal surface of the posterior articular surface in RBCM P900 is more strongly pointed, and overlaps the preceding phalanx more extensively, than in other small ornithischians with ginglymoid phalanges from similar stratigraphic and geographic ranges, such as parksosaurids (e.g., Parksosaurus ROM 804) and pachycephalosaurids (e.g., Stegoceras UALVP 2). Ginglymoid articular surfaces, and narrow, pointed unguals, also exclude identifications of this specimen as a juvenile ceratopsid (e.g., Chasmosaurus UALVP 52613, Currie et al. 2016) or hadrosaurid (e.g., Edmontosaurus annectens, LACM 23504 (Prieto-Márquez, 2014), RAM 7150 (Zheng, Farke & Kim, 2011), Lambeosaurinae indet., TMP 1998.058.0001). The relatively long and robust forelimb compared to the hindlimb, as indicated by the proportions of the radius and tibia, exclude RBCM P900 from being assigned to Thescelosauridae and Pachycephalosauria. The preserved elements of RBCM P900 are comparable in size to large leptoceratopsid specimens like Cerasinops MOR 300 and Leptoceratops CMN 8889.

Pectoral girdle

Arbour & Graves (2008: Figs. 2G and 2H) were unable to identify a thin, gently curved element of RBCM P900, which we reinterpret here as a fragmentary right coracoid (Fig. 3A). Most of the edges are broken, but the angle of the sternal process is complete and part of the anterior edge is complete. The morphology of this bone compares well with the complete coracoids of Leptoceratops CMN 8889 (Fig. 3B); the coracoids of most other Laramidian leptoceratopsids are incomplete and cannot be compared with RBCM P900. As in Leptoceratops, RBCM P900 had a pronounced, sharply pointed sternal process at the anterior and ventral end of the coracoid. The anterior edge of the coracoid in RBCM P900 appears straighter compared to the more curved edge in CMN 8889 (Leptoceratops), but without comparable material from other taxa it is difficult to assess whether or not this is within the range of intraspecific variation or a taxonomic difference.

Figure 3 Pectoral elements of RBCM P900, holotype of Ferrisaurus sustutensis, compared to other Laramidian leptoceratopsids.

(A) Fragmentary right coracoid of RBCM P900 in lateral view, compared to (B) complete right scapulocoracoid of CMN 8889, Leptoceratops gracilis, lateral view centered on coracoid with scapula in oblique view. Fragmentary left scapular blade of RBCM P900 in (C) lateral and (D) medial view, compared to (E) left scapula of MOR 300, Cerasinops hodgskissi in medial view, and (F) left scapula of TCM 2003.1.9, Prenoceratops pieganensis in lateral view. Abbreviations: sp, sternal process.

A fragmentary flattened bone was interpreted as a possible rib by Arbour & Graves (2008: Figs. 2E and 2F) and is reinterpreted here as part of the left scapula (Figs. 3C and 3D), representing a section near the midpoint of the scapular blade. It has a teardrop-shaped cross section on one side and rapidly narrows to a thin oval cross-section on the other side. The teardrop-shaped outline at one end precludes identification of this element as a rib shaft, and ribs for this individual would have been much smaller and less robust, whereas the proportions are more in line with the scapula of a leptoceratopsid with hindlimb proportions of this size. The ventral edge of the fragment is straight, and the dorsal edge is markedly concave. The scapulae of Montanoceratops (MOR 452) and Prenoceratops (TCM 2003.1.9 and TCM 2003.1.11; Fig. 3F) are relatively straight along their dorsal lengths, whereas the scapulae of Cerasinops (MOR 300, Fig. 3E) and Leptoceratops (CMN 8889) are more concave dorsally in lateral view.

Forelimb

We agree with the identification of the radius by Arbour & Graves (2008: Figs. 2C and 2D). The radius is a relatively simple rod-shaped bone with gently expanded proximal and distal ends and a shaft that is triangular in cross section (Figs. 4A–4D; Table 1). Overall, the radius of RBCM P900 is very similar to that of Leptoceratops (Brown, 1914; Fig. 4E), and it differs only in subtle aspects. The proximal end in RBCM P900 is less cup-shaped compared to Leptoceratops (CMN 8889), and the shaft lacks the prominent protuberance present near the midpoint in Leptoceratops (AMNH 5205; Brown, 1914), although a light distal tuberosity is present as in AMNH 5205. The preserved radii of Cerasinops (MOR 300; Fig. 4F) lack distal and proximal ends, but preserve straight shafts lacking any bulges or tuberosities.

Figure 4 Radius of RBCM P900, holotype of Ferrisaurus sustutensis, compared to other Laramidian leptoceratopsids.

RBCM P900, Ferrisaurus sustutensis, left radius in (A) lateral, (B) medial, (C) proximal, and (D) distal view. (E) CMN 8889, Leptoceratops gracilis, left radius in lateral view. (F) MOR 300, Cerasinops hodgskissi, ?left radius in ?lateral view. Abbreviations: tb, tubercle.

Table 1 Selected measurements of forelimb and hindlimb elements in leptoceratopsids (mm).

Taxon	Specimen	Radius	Ulna	Tibia	Fibula	Measurement source	
Length	Length	Distal width	Length	Distal width	Length		
Ferrisaurus sustutensis	RBCM P900	135.0		38.2		90.1		Direct measurement	
Cerasinops hodgskissi	MOR 300 R		201.4	>32.6	363.0	~86.3	337.0	Direct measurement	
MOR 300 L					95.0		Direct measurement	
USNM 13863				200	62		Brown & Schlaikjer (1942)	
Ischioceratops zhuchengensis	ZCFM V0016				329			He et al. (2015)	
Leptoceratops gracilis	AMNH 5205	167	224			117		Sternberg (1951)	
CMN 8887	115			240			Sternberg (1951)	
CMN 8888	137			290			Sternberg (1951)	
CMN 8889 L	160.5	202.5	35.7	323.0	87.9	293.0	Direct measurement	
PU 18133				~385	~78		Ostrom (1978)	
Montanoceratops cerorhynchus	AMNH 5464				355	102		Brown & Schlaikjer (1942)	
MOR 542			28.6	249.2	50.6	235.9	Direct measurement	
Prenoceratops pieganensis	TCM 2003.1.8		143.3	19.1				Direct measurement	

We reinterpret the bone previously identified by Arbour & Graves (2008: Fig. 2) as the proximal half of a humerus as a partial right ulna including the distal end (Figs. 5A and 5B; Table 1). The absence of a prominent deltopectoral crest or rounded humeral head is inconsistent with its identification as a humerus. The ulna is incomplete proximally, but the shaft is expanded toward the broken proximal end. Based on the proportions of the radius length to ulna length in Leptoceratops, Montanoceratops, and to a lesser extent Cerasinops (Supplemental Information 2) where the radius is 75–80% of the length of the ulna, the ulna of RBCM P900 may have been 170–180 mm in total length. The proximal expansion of the ulna occurs approximately 100 mm from the base of this element in RBCM P900, compared to about 120 mm in CMN 8889 (Leptoceratops; about 59% of the total length from the base), 96 mm in TCM 2003.1.8 (Prenoceratops; 67% of the length), and 125 mm in MOR 300 (Cerasinops; 62% of the length). Extrapolating a total length for the ulna of RBCM P900 based on these proportions yields a total length of ~150–170 mm. Comparing the width of the distal ulna to the length of the radius, the ulna of RBCM P900 was proportionately wider compared to other leptoceratopsids (Fig. 5; Supplemental Information 2), giving it a stouter appearance.

Figure 5 Ulna of RBCM P900, holotype of Ferrisaurus sustutensis, compared to other Laramidian leptoceratopsids.

RBCM P900, Ferrisaurus sustutensis, left ulna in (A) medial and (B) distal view. (C) CMN 8889, Leptoceratops gracilis, left ulna in medial view. (D) MOR 300, Cerasinops hodgskissi, right ulna in medial view. (E) TCM 2003.1.8, Prenoceratops pieganensis, right ulna in medial view. (F) MOR 452, Montanoceratops cerorhynchus, right ulna in medial view. (G) RBCM P900, Ferrisaurus left ulna in posterior view; arrow indicates medial bend to distal ulna. (H) TCM 2003.1.8, Prenoceratops right ulna in anterior view. (I) MOR 452, Montanoceratops right ulna in anterior view.

The ulna shaft is a flattened oval in cross-section, and the distal end is flat and only moderately expanded. A diagnostic character for Cerasinops proposed by Chinnery & Horner (2007) is the strong medial bend of the distal part of the ulna. The distal ulna of RBCM P900 is also medially deflected (Fig. 5G), with the posterior edge more strongly curved than the anterior edge. The postcrania of the bonebed material of Prenoceratops was not previously described by Chinnery (2004), but examination of TCM 2003.1.8, a right ulna (Fig. 5H), indicates that Prenoceratops also had a medial bend to the distal ulna. The ulna is straight in this region in Leptoceratops (CMN 8889) and Montanoceratops (MOR 542; Fig. 5I).

Hindlimb

Approximately the distal two thirds of the right tibia and fibula are preserved, with the tibia and fibula in articulation (Figs. 6A–6E; Table 1). Using more complete specimens of similar size as a guide (Supplemental Information 2), we estimate that the tibia in RBCM P900 was likely between 310 and 330 m in length originally. The astragalus and possibly the calcaneum are coossified to the tibia (Fig. 6C) but the boundaries between these elements are difficult to discern. The tibia and astragalus are not coossified in Leptoceratops (CMN 8889; Figs. 6H and 6I), Cerasinops (MOR 300; Figs. 6J–6L) or Montanoceratops (MOR 542) and in these specimens the boundary between these elements is clearly discernible. Makovicky (2010) notes that the astragalus is partly coossified with the tibia in Montanoceratops (AMNH 5465). It is unclear whether this is an ontogenetic phenomenon, and if it is phylogenetically significant.

Figure 6 Tibia of RBCM P900, holotype of Ferrisaurus sustutensis, compared to other Laramidian leptoceratopsids.

RBCM P900, Ferrisaurus left tibia in (A) medial, (B) and (C) posterior, (D) anterior, and (E) lateral views, and (F) and (G) block removed from anterior face of tibia containing four partial metatarsals. The dashed line in (C) delineates the possible boundary of the astragalus/calcaneum on the tibia, and the dashed lines in (E) indicate the preserved metatarsals in cross-section. CMN 8889, Leptoceratops gracilis left tibia in (H) posterior and (I) anterior view. MOR 300, Cerasinops hodgskissi right tibia in (J) anterior and (K) posterior views, and (L) left tibia in posterior view. Abbreviations: as, astraglus; ca, calcaneum; fib, fibula; ma, matrix; mt, metatarsal.

In medial and lateral views (Figs. 6A and 6E) the tibia of RBCM P900 has a pronounced distal curvature that was not observed in any other leptoceratopsid specimens and which does not seem to represent taphonomic deformation, based on the absence of crushing or fractures on the tibia. In distal view (Fig. 6B), the lateral and medial malleoli are offset at a distinct angle, giving the distal face of the tibia/astragalus a triangular cross section; RBCM P900 has a more pronounced edge marking the confluence between the malleoli compared to the condition in Leptoceratops (CMN 8889), Cerasinops (MOR 300), or Montanoceratops (MOR 542). The tibia of RBCM P900 is straight-sided in anterior and posterior view and tapers toward the midpoint in anterior or posterior view, similar to the condition in Leptoceratops (CMN 8889; Figs. 6H and 6I), and Montanoceratops (MOR 542), and unlike the strongly kinked morphology observed in Cerasinops (MOR 300; Figs. 6J–6L). The tibia narrows significantly along the shaft and has an oval cross section at its broken proximal end. The fibula is narrow, with an oval cross section. A portion of matrix removed from the anterior side of the distal tibia contains what appear to be the remains of four metatarsals in cross section (Figs. 6F and 6G), but little can be said about their morphology without further preparation.

RBCM P900 preserves a large number of pedal phalanges: III-2, III-3, and III-4, and IV-2, IV-3, IV-4, and IV-5 (Figs. 7A–7C; Table 2). Pedal digit III was preserved in articulation on a piece of matrix (Figs. 7A and 7B); digit IV includes IV-2 and IV-3 preserved in articulation and IV-4 and IV-5 can be “snapped” back into articulation based on the presence of some remaining matrix on these elements (Fig. 7C). The non-ungual phalanges are somewhat longer than wide, but blocky rather than elongate, and ginglymoid, distinguishing them from similarly-sized small-bodied ornithischians such as Parksosaurus (Fig. 7I). The distinctly ginglymoid nature of the interphalangeal joints is distinct from the non-ginglymoid pedal phalangeal joints in Hadrosauridae (Zheng, Farke & Kim, 2011).

Figure 7 Pedal elements of RBCM P900, holotype of Ferrisaurus sustutensis, compared to other Laramidian small-bodied ornithischians.

RBCM P900, Ferrisaurus, left digit III in (A) medial and (B) lateral views, and (C) left digit IV in lateral view. (D) MOR 542, Montanoceratops cerorhynchus, right digit IV in lateral view. Illustrations of (E) RBCM P900, Ferrisaurus sustutensis, (F) CMN 8889, Leptoceratops gracilis, (G) MOR 300, Cerasinops hodgskissi, (H) MOR 542, Montanoceratops cerorhynchus, and (I) ROM 804, Parksosaurus warreni, in dorsal view.

Table 2 Lengths of phalanges from pedal digits III and IV in leptoceratopsids (mm).

Taxon	Specimen	III	IV	Measurement source	
2	3	4	2	3	4	5		
Ferrisaurus sustutensis	RBCM P900	28.1	28.3	40.7	24.4	21.1	22.3	29.3	Direct measurement	
Cerasinops hodgskissi	MOR 300 R	?34.1	?27.9			?25.4			Direct measurement	
MOR 300 L	36.5	29.8	44.9	33.5	27.2	20.1	>31.1	Direct measurement	
USNM 13863	27.5	29.5	41	21	21	18.5		Brown & Schlaikjer (1942)	
Leptoceratops gracilis	CMN 8887	21.3	16.5	32.2	18.4	14.4	13.7	24.9	Direct measurement from cast	
CMN 8889 R	31.9	29.6	51.0	29.4	25.1	22.0	44.1	Direct measurement	
PU 18133	40	30		32	~26	~20		Ostrom (1978)	
Montanoceratops cerorhynchus	AMNH 5464	33		68					Brown & Schlaikjer (1942)	
MOR 542	28.4	25.1	~29.0	20.4	21.0	19.1	34.0	Direct measurement	

In Leptoceratops (CMN 8889, CMN 8887), Cerasinops (MOR 300), and Montanoceratops (MOR 542, Fig. 7D) the penultimate pedal phalanx of each major digit is markedly shorter in length compared to the preceding phalanx (~75–90% the length of the preceding phalanx); in RBCM P900 the penultimate and preceding phalanx on digits III and IV are similar in size, with the penultimate phalanx actually being slightly longer than the preceding phalanx (Table 2; Supplemental Information 2). Leptoceratops (AMNH 5205; Brown, 1914) and Cerasinops (USNM 13863; Gilmore, 1939; Chinnery & Horner, 2007) are both illustrated with penultimate phalanges subequal in length to the preceding phalanx, but these are both illustrated as line drawings rather than photographs, measurements were not provided by the authors, the digits in AMNH 5205 were not part of an articulated pes, and neither of these specimens were measured for this study. As such, it is unclear if the illustrations accurately reflect the actual morphology of the pedal digits in these two specimens. The figured pes of Udanoceratops PIN 4046/11 (Tereschenko, 2008) appears to show penultimate phalanges subequal in length to the preceding phalanx in digits II-IV, but measurements were not provided, and no rationale was provided for why this specimen is referred to Udanoceratops rather than Protoceratops. RBCM P900 can, however, be differentiated from Udanoceratops by the morphology of the pedal unguals, if PIN 4046/11 (Tereschenko, 2008) is referable to Udanoceratops rather than Protoceratops.

The unguals of RBCM P900 are long and narrow, with a gently curved ventral surface (Figs. 7A–7C and 7E), differing from the broad, hoof-shaped unguals of ceratopsids or the wide triangular unguals of protoceratopsids (Sternberg, 1951). Their overall shape is similar to the unguals of most other leptoceratopsids, with the possible exception of Udanoceratops based on specimen PIN 4046/11 where the proximal articular surface of the ungual is much wider than the distal articular surface of the penultimate phalanx (Tereschenko, 2008). Lateral grooves on the unguals of RBCM P900 are shallow. The unguals of Leptoceratops specimen CMN 8889 have a longitudinal furrow on the ventral surface, but these are absent in the smaller Leptoceratops specimen CMN 8887, and ventral furrows were not observed on any other leptoceratopsid unguals examined for this study. No ventral furrows are present on the unguals of RBCM P900. The unguals of RBCM P900 appear slightly deeper in lateral view compared to other leptoceratopsids, but it is unclear how much this is influenced by taphonomic factors (e.g., the pedal elements of Cerasinops MOR 300 are severely crushed), ontogeny, or body size (e.g., Montanoceratops MOR 542 is substantially smaller than RBCM P900).

Results of the phylogenetic analyses

The phylogenetic analysis of the He et al. (2015) modified matrix recovered seven most parsimonious trees, each with a tree length of 328, a consistency index of 0.60, a retention index of 0.80, and a best tree-bisection reconnection score of 326 (Fig. 8A). The strict consensus tree (Fig. 8) is nearly identical to that presented by He et al. (2015), with a basal grade of small-bodied ceratopsians and two derived clades, Coronosauria and Leptoceratopsidae. Within Leptoceratopsidae, the recovered relationships are similar to those found by He et al. (2015), with Asiaceratops and Cerasinops recovered as successive sister taxa to all other leptoceratopsids, Montanoceratops and Ischioceratops as sister taxa, and Prenoceratops as the sister taxon to an unresolved clade of the six remaining leptoceratopsids, including Ferrisaurus. Within this clade, Ferrisaurus has an unresolved relationship with the North American taxa Leptoceratops, Gryphoceratops and Unescoceratops and the Asian taxa Udanoceratops and Zhuchengceratops. Poor resolution of this group is most likely because of the low number of characters that could be coded for Ferrisaurus. In two of the seven trees, Ferrisaurus and Udanoceratops were sister taxa; the position of Ferrisaurus differs in the other five trees. Moving Ferrisaurus basally in the tree, outside of Coronosauria + Leptoceratopsidae, increases the tree length to 329, and moving Ferrisaurus into Ceratopsidae increases the tree length to 331.

Figure 8 Results of the phylogenetic analyses.

Strict consensus trees showing the relationships of Ferrisaurus sustutensis within Ceratopsia: (A) Strict consensus tree using the matrix modified from He et al. (2015). (B) Strict consensus tree using the matrix from Morschhauser et al. (2019).

The analysis of the Morschhauser et al. (2019) unmodified matrix recovered 1,110 most parsimonious trees, each with a tree length of 694, a consistency index of 0.45, a retention index of 0.67, and a best tree-bisection reconnection score of 688. Morschhauser et al.’s (2019) strict consensus tree shows a poorly resolved sister clade to Coronosauria consisting of taxa typically recovered as leptoceratopsids in other analyses plus Koreaceratops and Helioceratops (Fig. 8B). The addition of Ferrisaurus to this matrix collapses this clade, and Ferrisaurus is recovered in an unresolved polytomy of leptoceratopsids plus Aquilops, Archaeoceratops, Auroraceratops, Helioceratops, and Koreaceratops, outside of Coronosauria. In 63% of the trees, Ferrisaurus is recovered as a leptoceratopsid in an unresolved clade consisting of Cerasinops, Ischioceratops, Leptoceratops, Montanoceratops, Prenoceratops, Udanoceratops, Zhuchengceratops, and Gryphoceratops + Unescoceratops, with Helioceratops as the outgroup.

Discussion

The fact that RBCM P900, the first dinosaur specimen recovered from the Sustut Basin, is a leptoceratopsid rather than one of the more commonly encountered groups in many coeval formations in western North America, such as hadrosaurs, ceratopsids, or tyrannosaurs, is surprising, especially given well-documented preservational biases against small-bodied dinosaurs in more fossiliferous areas (Brown et al., 2013a, 2013b; Evans et al., 2013). Most leptoceratopsid taxa are distinguished on the basis of cranial morphology, especially aspects of the lower jaw anatomy (Ryan et al., 2012). However, excellent postcranial material is known for many taxa, making it possible to identify diagnostic features in RBCM P900 despite the absence of cranial material for this specimen. Leptoceratops, Montanoceratops, and Cerasinops are all known from multiple partial or complete skeletons (Chinnery & Weishampel, 1998; Chinnery & Horner, 2007; Ostrom, 1978; Brown & Schlaikjer, 1942; Sternberg, 1951; Brown, 1914), and Prenoceratops specimens described by Chinnery (2004) come from a single mixed bonebed from which multiple composite skeletons have been assembled.

Digit proportions have been used to distinguish caenagnathids (Zanno & Sampson, 2005), oviraptorids (Longrich, Currie & Zhi-Ming, 2010), and ornithomimids (Kobayashi & Barsbold, 2006) at low taxonomic levels, and we show that they can also be used to distinguish among leptoceratopsids. In all specimens preserving partial or complete articulated pedes that we were able to personally observe and measure, the penultimate phalanx (preceding the ungual) for each major digit is shorter in length than the immediately preceding phalanx. In other words, pedal phalanx length decreases distally in the digit, except for the unguals (Fig. 7). In Ferrisaurus, the penultimate phalanx is subequal in length to the preceding phalanx in digits 3 and 4, and phalanx length does not decrease distally within each pedal digit. This appears to be unique to Ferrisaurus within leptoceratopsids with two possible exceptions. This morphology may be present in a referred specimen of Udanoceratops (PIN 4046/11, Tereschenko, 2008), although it is not clear that this specimen is not referable to Protoceratops, and measurements were not provided. Gilmore (1939) published measurements for USNM 13863 (Cerasinops) and noted the length of III-2 as 27.5 and III-3 as 29.5 mm; although we have not had the opportunity to observe this specimen in person, a two mm length increase between III-2 and III-3 is far outside the range of variation we observed in leptoceratopsids over the course of this study (Table 2; Supplemental Information 2), but is within the range of variation of a decrease in length between III-2 and III-3. Additionally, phalanges in digit IV show the more typical reduction in length distally. We suggest it is possible that III-2 and III-3 in USNM 13863 were at some point transposed in their positions, despite the pes being reported as articulated at the time of collection by Gilmore (1939). Longer penultimate phalanges may also be present in more basal ceratopsian taxa such as Archaeoceratops (Dodson, 2003), although phalangeal measurements were not provided in the descriptions of this taxon; phalangeal length decreases in Yinlong as for leptoceratopsids except Ferrisaurus (Han et al., 2018). Overall, the observed pattern for leptoceratopsids appears to be a marked decrease in non-ungual phalangeal length in each pedal digit, with the exception of Ferrisaurus.

The astragalus and tibia in RBCM P900 are coossified (Fig. 6), an unusual condition among leptoceratopsids that is otherwise reported in only one specimen of Montanoceratops (AMNH 5465). Coossification of the astragalus and tibia could indicate advanced skeletal maturity in RBCM P900, but this specimen is smaller than specimens in which the tibia and astragalus remain separate (e.g., Leptoceratops CMN 8889, Montanoceratops AMNH 5205), suggesting that size alone does not explain the differences in coossification patterns in leptoceratopsids. It is unclear what the ontogenetic significance of this coossification represents in Ferrisaurus. Fusion of the ankle (distal tibia and fibula) has been proposed as a diagnostic character of the small bodied thescelosaurid Albertadromeus syntarsus from the Campanian of Alberta (Brown et al., 2013b), and a distinctive feature of some coelophysoids such as “Syntarsus”/?Coelophysis kayentakatae (Rowe, 1989), mature derived ceratosaurs, such as Cryolophosaurus (Smith et al., 2007), and mature ankylosaurs (Coombs, 1971) and ceratopsids (Sues & Averianov, 2009).

Ferrisaurus shares with Cerasinops a medially bent distal ulna (originally proposed as a diagnostic character for Cerasinops by Chinnery & Horner, 2007), a feature that is also present in Prenoceratops (TCM 2003.1.8). This feature is not present in the Maastrichtian-aged leptoceratopsids Montanoceratops and Leptoceratops, which are closest in geological age to Ferrisaurus. Chinnery & Horner (2007) suggested that the medial deflection of the ulna in Cerasinops, as well as the proportions and histology of the limb elements, may indicate that Cerasinops was primarily bipedal rather than quadrupedal. Although limb proportions are more difficult to determine in Ferrisaurus, if the complete tibia was between 310 and 330 mm (estimated based on more complete tibiae in Leptoceratops and Montanoceratops, Supplemental Information 2), then the radius of Ferrisaurus would have been no more than 40–43% of the length of the tibia. This is less than other comparable leptoceratopsids: the radius is 50% the length of the tibia in Leptoceratops CMN 8889, 48% in Leptoceratops AMNH 5205, and 47% in Leptoceratops CMN 8888, and much more than 45% in the incomplete radii of Cerasinops MOR 300. Ferrisaurus thus may have had a more robust distal ulna (Fig. 5), but a shorter forelimb overall compared to Cerasinops, suggesting that it too may have been at least facultatively bipedal. Alternately, the robusticity of the ulna may be related to another aspect of its ecology, such as digging, which has been suggested in the orodromine Oryctodromeus (Fearon & Varricchio, 2015) and Protoceratops (Longrich, 2010).

Ferrisaurus was recovered as a leptoceratopsid using the modified He et al. (2015) character matrix, and as a non-coronosaurian neoceratopsian in the Morschhauser et al. (2019) matrix. Although the precise relationships of Ferrisaurus are unresolved using the He et al. (2015) matrix, we found it to be more closely related to Leptoceratops than Montanoceratops (Fig. 8). Despite their stratigraphic and geographic proximity, Leptoceratops and Montanoceratops are not recovered as close relatives in recent phylogenetic analyses in this analysis or by He et al. (2015) and preceding versions of that matrix. Montanoceratops occupies a relatively basal position within Leptoceratopsidae (Makovicky, 2010; Ryan et al., 2012; Farke et al., 2014; He et al., 2015), and was found to be the sister taxon to Ischioceratops from Asia by He et al. (2015). Leptoceratops typically occupies a more derived position and has been recovered as the sister taxon to the Asian Udanoceratops (He et al., 2015). Ferrisaurus was thus recovered in a more derived position within Leptoceratopsidae relative to Montanoceratops.

Stratigraphic and palaeobiogeographic implications

Leptoceratopsids are known from the Santonian through Maastrichtian of Laramidia (Ryan et al., 2012), and the Campanian–Maastrichtian of Mongolia and China (He et al., 2015); fragmentary putative leptoceratopsids have also been reported from the Cenomanian of Uzbekistan (Nessov, Kaznyshkina & Cherepanov, 1989), the ?Coniacian-Santonian of Belgium (Godefroit & Lambert, 2007; Longrich, 2016), the Campanian of North Carolina (Longrich, 2016), and the Campanian of Sweden (Lindgren et al., 2007). The ancestor of the leptoceratopsid lineage most likely originated in Asia (Chinnery-Allgeier & Kirkland, 2010), but multiple exchanges across Beringia from Asia to North America (and vice versa) may have occurred. Gryphoceratops, the oldest taxon, derives from the Deadhorse Coulee Member of the Milk River Formation, with a minimum age of about 83.7 Ma (Ryan et al., 2012). Campanian Laramidian taxa include Cerasinops from the lower Two Medicine Formation, Prenoceratops from the upper Two Medicine Formation of Montana and the Oldman Formation of Alberta, and Unescoceratops from the lower Dinosaur Park Formation (Chinnery, 2004; Chinnery & Horner, 2007; Ryan et al., 2012). Only two genera are known from the Maastrichtian of Laramidia: Montanoceratops from the St Mary River and Horseshoe Canyon formations (Brown & Schlaikjer, 1942; Chinnery & Weishampel, 1998; Makovicky, 2001), and Leptoceratops from the Scollard and Hell Creek formations (Sternberg, 1951; Ott, 2007) and the Pinyin Conglomerate (McKenna & Love, 1970). RBCM P900 was most likely collected from approximately 68.2–67.2 Ma sediments of the Tatlatui Member of the Tango Creek Formation, based on a recent field reassessment of its original collection locality and palynomorphs recovered from that site (Arbour et al. (in press)). This places it between the stratigraphic ranges for Montanoceratops (71.939–68 Ma) and Leptoceratops (66.97–66 Ma), and slightly overlapping with the known range of Montanoceratops (Fowler, 2017).

Stratigraphically, Montanoceratops and Leptoceratops are the most likely taxa to which RBCM P900 could be referred, but multiple anatomical features distinguish RBCM P900 from both Leptoceratops and Montanoceratops, including the proportions of the pedal digits, the proportions of the ulna, and the medially bowed morphology of the distal ulna. RBCM P900 is also unlikely to represent an individual of Cerasinops or Prenoceratops; it can be distinguished from Cerasinops based on the proportions of the pedal digits, and from both Cerasinops and Prenoceratops based on the proportions of the ulna. These morphological differences are reinforced by the stratigraphic position of Ferrisaurus relative to Cerasinops and Prenoceratops (latest Maastrichtian, vs. middle to Upper Campanian; Chinnery & Horner, 2007; Chinnery, 2004), given that no other dinosaur species with temporally well-resolved specimens spans the middle Campanian to latest Maastrichtian elsewhere in Laramidia (Eberth et al., 2013; Fowler, 2017). An enigmatic specimen, TMP 1982.11.1, from the Maastrichtian Willow Creek Formation (Miyashita, Currie & Chinnery-Allgeier, 2010) has been referred to Montanoceratops by several authors (Ryan & Currie, 1998), but was considered neither a representative of Montanoceratops, Leptoceratops, or Cerasinops by Makovicky (2010). Several additional as-yet undescribed specimens in the collections of the TMP (Tanke, 2007) may represent examples of either Montanoceratops, Leptoceratops, or Ferrisaurus and their description may help clarify the differences between these three taxa or provide new anatomical information for Ferrisaurus.

Leptoceratopsids are uncommon components of the dinosaurian faunas of Laramidia: even in the well-sampled Dinosaur Park Formation of Alberta only a handful of leptoceratopsid specimens are known (Tanke, 2007). Ryan & Evans (2005) hypothesized that leptoceratopsids may have avoided the wet coastal environments favored by ceratopsids. Leptoceratops appears to be present primarily in piedmont and alluvial plain palaeoenvironments and is largely absent in coastal plain settings (Lehman, 1987, although see Ott, 2007). The Tatlatui Member of the Tango Creek Formation represents an alluvial plain palaeoenvironment (Bustin & McKenzie, 1989), consistent with the palaeoenvironmental association documented for other Maastrichtian leptoceratopsids. Interestingly, the intermontane basin occurrence of Ferrisaurus also supports one hypothesis outlined by Lehman (1987, 2001), that leptoceratopsids, along with a few other large-bodied herbivorous taxa, were inhabitants of Cordilleran highlands and adjacent piedmonts, which, in part, explains their rarity in the fossil record.

Although today the holotype locality for Ferrisaurus is found at approximately 56°N, the unusual and complex translational history of the Intermontane Superterrane means its palaeolatitude may have lain as much as 1,600 km to the south of its current position with respect to cratonic North America, and may have had approximately the same palaeolatitude (~48°N) as the southern border of Oregon and Idaho (Enkin et al., 2003; Van Hinsbergen et al., 2015). Despite its current apparent northern latitude, the holotype of Ferrisaurus may actually represent one of the southernmost occurrences of Leptoceratopsidae in western North America, and at minimum would have been within the currently known latitudinal range of Laramidian leptoceratopsids. Regardless, RBCM P900 represents a western range extension for Laramidian leptoceratopsids, and a unique occurrence within a restricted intermontane basin palaeoenvironment. The identification of RBCM P900 as a unique leptoceratopsid distinct from other known Laramidian taxa supports previous conclusions by Makovicky (2010) and Ryan et al. (2012) that Leptoceratopsidae was a diverse but currently poorly sampled lineage of Late Cretaceous ceratopsians.

Conclusions

RBCM P900, previously identified as an indeterminate bipedal neornithischian by Arbour & Graves (2008), instead represents the partial skeleton of a leptoceratopsid ceratopsian similar in size to large specimens of Leptoceratops and Cerasinops. Although fragmentary, this specimen can be differentiated from other leptoceratopsids based on the proportions and morphology of the ulna and pedal digits, and is designated the holotype of the new taxon Ferrisaurus sustutensis. RBCM P900 was collected from the Sustut Group of the southern Sustut Basin, a large but relatively unexplored terrestrial Cretaceous basin in northern British Columbia, Canada. Its recognition as a distinct species of a generally rare group of small-bodied dinosaurs highlights the potential for future discoveries of unique dinosaur biodiversity within the intermontane basins of the western side of the North American Cordillera.

Supplemental Information

Supplemental Information 1 Specimens examined and character statements.

Click here for additional data file.

Supplemental Information 2 Comparative measurements.

Click here for additional data file.

Supplemental Information 3 Character-taxon matrix.

Click here for additional data file.

Supplemental Information 4 Character-taxon matrix modified from Morschhauser et al. (2019).

Click here for additional data file.

The RBCM P900 field locality is located on the unceded traditional territory of the Gitxsan peoples. MOR 300 was collected from the Wilson Hodgkiss Ranch, and MOR 542 was collected from private land deeded from the Blackfeet Nation. Many thanks to Dallas Evans (Children’s Museum of Indianapolis), Jordan Mallon and Kieran Shepherd (Canadian Museum of Nature), Amy Atwater, Scott Williams, and John Scannella (Museum of the Rockies), Brandon Strilisky and Caleb Brown (Royal Tyrrell Museum of Palaeontology), and Carl Mehling (American Museum of Nature History) for access to specimens in their collections. Peter Makovicky shared photographs of Udanoceratops and Montanoceratops, Kentaro Chiba, Cary Woodruff and Bobby Boessenecker provided assistance with digital modelling and photogrammetry, and Derek Larson provided assistance with Latinization of the genus name. Many thanks to editor Hans-Dieter Sues and reviewers Andy Farke and Brenda Chinnery for constructive comments that improved the manuscript.

INSTITUTIONAL ABBREVIATIONS

LACM Los Angeles County Museum

MOR Museum of the Rockies, Bozeman, Montana, USA

RBCM Royal BC Museum, Victoria, British Columbia, Canada; Raymond M. Alf Museum of Paleontology, Claremont, California, USA

ROM Royal Ontario Museum, Toronto, Ontario, Canada

TMP Royal Tyrrell Museum of Palaeontology, Drumheller, Alberta, Canada

UALVP University of Alberta, Edmonton, Alberta, Canada

CMN Canadian Museum of Nature, Ottawa, Ontario, Canada.

Additional Information and Declarations

Competing Interests

Author Contributions

Data Availability

New Species Registration

The authors declare that they have no competing interests.

Victoria M. Arbour conceived and designed the experiments, performed the experiments, analyzed the data, contributed reagents/materials/analysis tools, prepared figures and/or tables, authored or reviewed drafts of the paper, approved the final draft.

David C. Evans conceived and designed the experiments, performed the experiments, authored or reviewed drafts of the paper, approved the final draft.

The following information was supplied regarding data availability:

Data is available in the Supplemental Files.

Specimens are accessioned at the following repositories:

Cerasinops hodgskissi, MOR 300, Museum of the Rockies, Bozeman, Montana, USA.

Ferrisaurus sustutensis, RBCM P900, Royal BC Museum, British Columbia, Canada.

Leptoceratops gracilis, CMN 8889, CMN 8887, Canadian Museum of Nature, Ottawa, Ontario, Canada.

Montanoceratops cerorhynchus, MOR 425, Museum of the Rockies, Bozeman, Montana, USA.

Prenoceratops pieganensis, TCM 2003.1.6, TCM 2003.1.2, TCM 2003.1.11, TCM 2003.1.5, TCM 2003.1.1, TCM 2003.1.4, TCM 2003.1.12, TCM 2003.1.3, TCM 2003.1.9, TCM 2003.1.7, TCM 2003.1.8, Children’s Museum of Indianapolis, Indianapolis, Indiana, USA.

The following information was supplied regarding the registration of a newly described species:

Publication LSID: urn:lsid:zoobank.org:pub:D1C60A34-3632-43AD-BCE0-C93D5E26D1B0.

Ferrisaurus LSID: urn:lsid:zoobank.org:act:8430CA06-567E-45A6-B91E-19E51502369E.

Ferrisaurus sustutensis gen. et sp. nov. LSID: urn:lsid:zoobank.org:act:A7F4267C-8CC6-49B6-8E52-2C2148929B14.

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
