# Peer review of "A new leptoceratopsid dinosaur from Maastrichtian-aged deposits of the Sustut Basin, northern British Columbia, Canada"

_PeerJ, doi:10.7717/peerj.7926_

## Round 0.1 · original submission · Major Revisions

Both reviewers offer a number of comments and suggestions that the authors should address when preparing a revised manuscript. Like one of the reviewers, I see no solid evidence for establishing a new taxon (which likely would become a nomen dubium very soon) and would recommend identification as Leptoceratopsidae gen. et sp. indet. This in no way lessens the importance of the find.

·

Basic reporting

The authors have made all of the necessary comparisons with other taxa, and the figures are generally quite helpful for evaluating their morphological descriptions. Overall, the text is clear and concisely written.

See general comments for additional suggestions and explanations.

Experimental design

The phylogenetic analysis should be expanded and updated. I would strongly advise including the specimen into a more comprehensive ornithischian phylogeny, such as that of Han et al. 2018, to test the placement of this fossil as a ceratopsian. Additionally, I would suggest using the Morschhauser et al. 2019 matrix either in addition to or in place of the current matrix. (I do not fault the authors on this latter point, because it was only *just* published on the week that I completed this review, and none of us had access to it prior to them). Finally, no matter which matrix is used, the robusticity in the placement of RBCM P900 should be explored through various constraint tests (e.g., constraining the fossil in different phylogenetic positions--how many extra steps to place it outside of Leptoceratopsidae?).

See general comments for additional suggestions and explanations.

Validity of the findings

The stated autapomorphy for RBCM P900 and its associated taxon related to phalangeal proportions is not supported as automorphic by the measurements in the supplemental data (e.g., for USNM 13863 and also CMN 8889, which comes close to approximating the same proportions), and should be explored in more detail. Based on the data in this paper, the distribution of the character seems to be broader than implied by the diagnosis (see explanation below), and is variable within a single taxon (e.g., CMN 8887 vs. CMN 8889). This point is critical for whether or not a new taxon should be erected, and must be addressed (in my opinion). At a minimum, I would suggest removing the distinction of "autapomorphy" and instead note that the taxon is distinguished by a unique combination of characters (rather than any particular individual autapomorphy). A more drastic change would be to simply designate the material as Leptoceratopsidae indet.

See general comments for additional suggestions and explanations.

Additional comments

This manuscript proposes a new identification for a fossil (RBCM P900) from the latest Cretaceous of British Columbia, suggesting that it is a leptoceratopsid. This finding is important both because the Cretaceous vertebrate body fossils of British Columbia are so poorly known, but also because leptoceratopsids in general are rather sparse in the fossil record. In my opinion, the assignment to Leptoceratopsidae seems reasonable, although I think that both this point as well as the diagnosis of the new taxon need some additional exploration and clarification. I am not yet confident that this can (or should) be given a new name, primarily because the stated autapomorphy around phalangeal proportions is apparently shared with other leptoceratopsids. Excluding this autapomorphy, there still is a presumably unique combination of characters, but given the fragmentary nature of the material and the uncertainty over intraspecific variation for other leptoceratopsids, I advise caution. This is of course a matter of personal preference, but at the least I think that the manuscript should be updated to address this and other issues outlined below.

General points include:

1) The phylogenetic analysis should be expanded and updated. I would strongly advise including the specimen into a more comprehensive ornithischian phylogeny, such as that of Han et al. 2018, to test the placement of this fossil as a ceratopsian. Additionally, I would suggest using the Morschhauser et al. 2019 matrix either in addition to or in place of the current matrix. (I do not fault the authors on this latter point, because it was only *just* published on the week that I completed this review, and none of us had access to it prior to them). Finally, no matter which matrix is used, the robusticity in the placement of RBCM P900 should be explored through various constraint tests (e.g., constraining the fossil in different phylogenetic positions--how many extra steps to place it outside of Leptoceratopsidae?).

2) The stated autapomorphy for RBCM P900 and its associated taxon related to phalangeal proportions is not supported as automorphic by the measurements in the supplemental data (e.g., for USNM 13863 and also CMN 8889, which comes close to approximating the same proportions), and should be explored in more detail. Based on the data in this paper, the distribution of the character seems to be broader than implied by the diagnosis (see explanation below), and is variable within a single taxon (e.g., CMN 8887 vs. CMN 8889). This point is critical for whether or not a new taxon should be erected, and must be addressed (in my opinion). At a minimum, I would suggest removing the distinction of "autapomorphy" and instead note that the taxon is distinguished by a unique combination of characters (rather than any particular individual autapomorphy). A more drastic change would be to simply designate the material as Leptoceratopsidae indet.

3) The authors have made all of the necessary comparisons with other taxa, and the figures are generally quite helpful for evaluating their morphological descriptions. Overall, the text is clear and concisely written.

SPECIFIC POINTS
- line 77/78 -- full institutional name and location for RAM is Raymond M. Alf Museum of Paleontology, Claremont, California, USA
- for the photogrammetric models, can you provide more details on how they were constructed? How many photos? What settings? What version of Metashape? I don't know that you need a ridiculous amount of detail here, but some basic additional methods should be explained. If relevant, consider including images of the surface models as a figure or supplemental figure; because these models strip away color, it can be helpful for showing morphology (strictly optional, but might be particularly helpful for showing the pedal phalangeal anatomy).
- Consider using the new Morschhauser et al. phylogenetic matrix for ceratopsians, also -- it is more recent, and includes a wide array of characters not included in the He et al. and Farke et al. matrices. Also, although I recognize that at some point you have to cut off the citation chain, the Farke et al. matrix was built on some previous matrices, including those of Ryan and Makovicky (the latter being the primary "grandparent" matrix of all of these), and those should be cited here.
- line 136 -- in general, I do not advocate citation of in review/unpublished papers; maybe just include as "Arbour et al., unpublished data"? Recognizing that you will include more detail in the other paper, I would still include a brief bit of info on how you reassigned the stratigraphic level--was it based on photos? Sketches? Distance from a local landmark? Again, I don't think you need to include comprehensive detail, but because the stratigraphic level is so important here, it is worth providing at least an additional sentence of explanation.
- where element identifications change from those previously published (e.g., the ulna from the previous interpretation of a humerus), please include some justification for why it is a better fit for one element than the other. E.g., morphological features that don't match, etc. This doesn't need a huge amount of text, but should be justified at least briefly.
- As shown in Figure 7H, the penultimate phalanx and preceding phalanx of digits III and IV appear subequal in MOR 542; can this really be used as an autapomorphy for the BC leptoceratopsid then?
- Gilmore 1939 (p. 2) indicates that the pes was articulated when found. This is alluded to in the description (lines 246/247), but should be more explicitly noted. The phalangeal proportions are so critical for this study, that I think it is very important to have some more information on USNM 13863. I have checked my files and don't have any photos, but it is possible that someone else out there might (e.g., Eric Morschhauser or Brenda Chinnery). I do not mandate an in-person visit to the fossil, but I would strongly advise getting more data on this, because it is so critical to the diagnosis and phylogenetic placement. I agree that the AMNH Leptoceratops gracilis holotype is probably not useful given the disarticulated and mixed nature of the material.
- The phylogenetic placement of this specimen should be explored in much more detail. For the ceratopsian analysis, how many steps does it take to move this around Ceratopsia? How robust is its placement as a leptoceratopsid? Are there any synapomorphies supporting this? Also, I would strongly suggest including the specimen into one of the general Ornithischia matrices (e.g., that of Han et al. 2018)--how robust is the placement in Ceratopsia? There won't likely be much in the way of synapomorphies, but I would be interested to see how robust the placement is.
- This is optional, but I would suggest including some basic line drawings showing the pes of the RBCM P900 versus those for Orodromeus, Parksosaurus, Stegoceras, etc., to emphasize the morphological distinctiveness of RBCM P900 from those other taxa.
- Lines 129/130; 229/230; etc. -- Is there *any* chance of additional preparation on those metatarsals? I don't know the matrix or the potential mount of additional material or the logistical feasibility, but given the taxonomic ambiguity of this material, there could (but no guarantee) be important features in this part of the skeleton.
- Although some measurements were included in the previous paper about this specimen, a measurement table would be very helpful here too. I see the supplemental info with this, but at least selected measurements should be in the main body of the manuscript. The supplemental information should be referred to in the text, also. Finally, I would advise reporting measurements only to the nearest 0.1 mm at most precise; the nearest millimeter may even suffice, given limitations in measuring these bones.
- Can you include an interpretive drawing showing the extent of the astragalus relative to the tibia? It's unclear from the photos.
- lines 308-309: Han et al. (2018) provide measurements of the pedal phalanges of Yinlong in the supplemental information, which may be useful here.
- lines 335-342: I don't put a lot of stock into the positions of particular taxa within Leptoceratopsidae; it doesn't take many steps to rearrange the tree (which is frustrating to me, but the reality of current matrices)
- line 411: I liked seeing the acknowledgment of traditional territorial boundaries. I hope this sets a positive precedent within our field!

FULL DISCLOSURE: I had a brief online conversation about aspects of phylogenetic matrices probably relevant to this project with the senior author. It does not, in my opinion, substantively affect my ability to evaluate this manuscript.

CITATIONS
Fenglu Han, Catherine A. Forster, Xing Xu & James M. Clark (2018) Postcranial anatomy of Yinlong downsi (Dinosauria: Ceratopsia) from the Upper Jurassic Shishugou Formation of China and the phylogeny of basal ornithischians, Journal of Systematic Palaeontology, 16:14, 1159-1187, DOI: 10.1080/14772019.2017.1369185

Eric M. Morschhauser, Hailu You, Daqing Li & Peter Dodson (2018) Phylogenetic history of Auroraceratops rugosus (Ceratopsia: Ornithischia) from the Lower Cretaceous of Gansu Province, China, Journal of Vertebrate Paleontology, 38:sup1, 117-147, DOI: 10.1080/02724634.2018.1509866

·

Basic reporting

Professional language is used throughout, with a few errors (please see annotated copy).
The introduction and background do show context and importance of the specimen, although if there is any way to strengthen the argument for the specimens coming from the site discussed, it would help. This is not an argument for rejecting the paper, as sometimes site information is not available, just a comment. The specimens were found in 1971 and were not discussed until 2008. Where were they during these years? The authors state that the original discoverer had field notes that were used to find the probable site of discovery, but only a turtle shell fragment possibly corroborates that this is the area where the specimens were found. Maybe the authors could share some of the original field notes?
The citations are for the most part fine, although when more than one reference is cited the authors did not put them in alphabetical order at least sometimes (lines 294-296 for example). Also some areas definitely need more citations, for example the stratigraphic and palaeobiogeographic implications section. Each taxon listed with an age or formation needs a citation. I’ve also made notes on the annotated copy.
The figures are great, very clear. Please double check the scale bars (see below), and I have some specific comments on the figures.
Figure 1 needs clarification on a continental scale, for those of us who don’t live in Canada.
Figure 2 is nice, I like it. The other figures are also great, although the difference in size of the radii in Figure 4 is surprising (please see the “Validity of the findings” section).
Raw data is supplied and is fine, although I think a different format than an excel file would be more user friendly.
Custom checks: I agree that this is most likely a new species of ceratopsian dinosaur, if for no other reason that it is the only specimen found in this area of the continent. It is correctly described and meets the ICZN standard. The autapomorphies in the diagnosis could use some work; one is presence of coossification of elements, which may not be diagnostic (especially when found in another species!), one is reliant on two elements being from the same specimen, which I am not sure of (see “Validity of the findings” section), and one is stated as “similar to” the condition found in two other species.

Experimental design

Research is within the scope of the journal.
No experimentation was done, methods seem sound, although direct measurements of elements were not described in terms of orientation of specimens or other testable criteria. Cladistic methods are also sound, but I don’t agree with some of the characters used in the analysis.
Many characters in this (and other) cladistic analyses are quite subjective, including words like “short” or “shorter” and “less” or “more”; these analyses are difficult to replicate, because of opinion as to what the words mean. It would be better to use percentages or other numerical data to differentiate character states, so that everyone who uses the character matrix is in agreement. However, I realize that I am in the minority on this issue, and the matrix in this manuscript is perfectly acceptable to most.
I caution the inclusion of all characters of the head (which is the vast majority of characters) because this new specimen has no cranial elements included with it.
It would be interesting to run a cladistics analysis with only the characters present on the new specimen, to see what the results of that are.

Validity of the findings

The specimens are definitely valid as possible elements of a new non-ceratopsid neoceratopsian. The fact that they are so fragmentary does not argue against their validity (I might be in the minority with that opinion), because it is unlikely that more specimens will be found from the area. I regret not describing the postcranial skeleton of Prenoceratops before the specimens were scattered across the globe. It would help this and other papers and projects. One problem with the Prenoceratops material is that there is quite a bit of individual variation among the elements. For example, fusion between elements is variable; one scapula and coracoid are fused, while the others are not. Fusion of elements, as discussed by these authors (and me previously), is not for sure an indication of maturity. I don’t personally think that it is taxonomically valid, either, and if this is the case then the authors should retract that character from their cladistic analysis.

I have doubts about the bone fragments all belonging to the same fossil. The RBCM P900 radius is much shorter than that of CMN 8889. In fact, I’m not sure that the scale is correct for Figure 4. The length measurement listed in Table 2 for the RBCM radius is 134.9mm, which is 84% of the length of the CMN radius (as listed in the same table). When I measure directly from the figure, the RBCM radius is 74% of the length of the CMN radius, although this could be due to scaling issues on my part. Regardless, the radius is much shorter and is also a different color (I know, could mean nothing), but the partial ulna with RBCM certainly appears to be similar in size, if not larger, than that of CMN 8889. If the authors do include this element with the specimen, then I believe it merits a discussion of what it means that it is so much shorter than that of the CMN specimen.

---

## Round 0.2 · accepted · Accept

The revised version of the manuscript will be recommended for acceptance for publication.